# Proposal of a Mathematical Model to Monitor Body Mass over Time in Subjects on a Diet

**DOI:** 10.3390/nu14173575

**Published:** 2022-08-30

**Authors:** Jose M. Soriano, Giovanna Sgambetterra, Pietro Marco Boselli

**Affiliations:** 1Food & Health Lab, Institute of Materials Science, University of Valencia, 46980 Valencia, Spain; 2Joint Research Unit on Endocrinology, Nutrition and Clinical Dietetics, University of Valencia-Health Research Institute La Fe, 46026 Valencia, Spain; 3Departament de Biosciencies, University of Milan, 20122 Milan, Italy

**Keywords:** BMI cut-offs, BMI, body fat, pseudothickness, obesity, BMI-BFMNU, anthropometric indices

## Abstract

Nowadays, slimming diet methodology works within a reduction of body mass using a decrease of dietary energy intake. However, there is no suitable method for understanding the dynamic process of body mass metabolic transformation over time. In the present paper, we have developed a biomathematic model to explain the temporal trend of body mass and its variations of people who have undergone a change in their diet using the solving equation of the model. Data relating to sex, age, body mass, and BMI were collected, and the compartmental model used to interpret the body mass trends was constructed by assuming that the mass results from the sum of the metabolic processes: catabolic, anabolic, distributive. The validation of the model was carried out by variance analysis both on the total and individual data sets. The results confirm that the trend of body mass and its variations over time depends on metabolic rates. These are specific to each individual and characterize the distribution of nutritional molecules in the various body districts and the processes catabolic, anabolic, distributive. Body mass and its variations are justified by the metabolic transformations of the nutritional quantities. This would explain why energetically equal diets can correspond to people of different body mass and that energy-different diets can correspond to people of body mass at all similar.

## 1. Introduction

Every biological phenomenon, including the dynamic processes regarding the absorption, distribution, metabolism and excretion of molecules obtained from the diet, should be interpreted as a function of time. In fact, the rule applied in the slimming diet is derived from the estimation of the energy content of the lost weight but ignores the dynamic physiological adaptations that arise during the change of weight, which lead to changes both in the basal metabolic rate and in energy expenditure for physical activity. Several models have been developed to verify changes in body mass over time. They have incorporated metabolic adjustment and a nonlinear resting metabolic rate (RMR) term [1], the use of a one-dimensional differential equation [2], the application of a multiplier for the adaptation of dietary-induced thermogenesis [3], the calculation of a new parameter that considers the energy expended to store energy during weight gain [4], the development of a differential equation model based on the first law of thermodynamics that incorporates all three adjustments along with the natural age-related reduction in the resting metabolic rate [5] and mechanism-based model spanning full individual life and capturing changes in body weight, composition and height [6]. However, no suitable method exists to describe trends in body mass over time; at best, a broken line fits individual instantaneous data related to mass, as in a discontinuous process. Therefore, interpolating and extrapolating data as a function of time is difficult because each situation was considered static and detached [5]. Predictive equations [7,8,9] combined with anthropometric parameters or not [10,11,12,13,14,15], have been applied to calculate the resting metabolic rate. It is therefore important to find a model whose solving equation can interpret the temporal trend of body mass in any environmental, physiological and/or pathological condition.

The aim of this study is to propose a mathematic model that explains trends in body mass and its variations using a continuous function, valid in any physiological and pathological condition.

## 2. Materials and Methods

For this study, we selected 374 patients of both sexes, aged between 7 and 73 years, with different pathophysiological conditions who came to our university laboratory, and at least four controls were performed during the nutritional pathway using the Biological-Physiological Model of Human Nutrition (BFMNU) method, which is a mathematical formula applicable in any physiological and pathological condition. The criterion was to motivate them to voluntarily join the research first of all to take advantage of the benefits of the new diet protocol, as well as to collaborate to improve the health service of prevention and treatment through nutrition. We decided to collect the data in a single broad set to stay consistent for the purpose of the work. Of course, the probability of finding a template for each of the endless categories of belonging of the people tested may be higher than finding one that applies to everyone. However, it would be better to find one of general meaning, valid for everyone regardless of age, sex, muscle activity and pathologies. The choice to consider only those who had had at least four check-ups was mandatory. In fact, checking a curvilinear pattern with two or three experimental points makes no sense. 

The number of experimental samples was adequate to calculate the statistical analysis. In order to obtain a dataset suitable for the analysis of variance, only subjects who had performed no less than four direct measurements of body mass over the entire transition period of initial body mass to the final one was admitted to the study. The size of the sample thus selected largely guaranteed the applicability of the statistical analysis. In fact, the distribution of variance indicates that a set of data pairs that has at least 240 degrees of freedom is congruous. Since only volunteers who had carried out at least four experimental checks would have been admitted to the study, 60 patients would have been enough (however, we worked with 374 patients). Anthropometric and weight measurements were carried out in the clinic by qualified personnel. The complete set of results was evaluated with variance analysis. This formula was considered an alternative to energy-based methods that enable the prediction of changes in body composition in response to energy intake through food, a crucial factor in the management of diet therapy. This method was devised by Boselli [16,17] with the objective of proposing a new nutritional intervention method different from the existing methods, which was able to systemically interpret the metabolic processes of mass and energy and to overcome the many and obvious contradictions of the methods already in use. The technical error of measurement (TEM), which is an accuracy index that represents the measurement quality and control dimension, was determined according to Perini et al. [18]. 

The Ethical Committee approved the study of the University of Milan (Milan, Italy) (Ethical Committee Number UnivMil-1-2014). All subjects provided written informed consent before study entry.

## 3. Results

The BFMNU model assumes that body mass results from the sum of three contributions from the catabolic, anabolic, and redistributive phases:

Catabolic phase
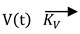

with K_V_ > 0
$$\frac{\mathrm{dV}\left(t\right)}{\mathrm{dt}}=-K_{V}V\left(t\right)$$
initial conditions:
V(0) = M_i_
final conditions:
V(t→∞) = 0
dV(t)/V(t) = −k_v_ dt; ∫ dV(t)/V(t) = −k_v_ ∫ dt; lnV(t) = −k_v_ t + cost.;
per t = 0 cost. = lnV(0) = lnM_i_; lnV(t) = lnM_i_ − k_v_ t;
lnV(t) − lnM_i_ = −k_v_ t; ln[V(t)/M_i_] = −k_v_ t; V(t)/M_i_ = e^− k^_v_^t^;
M_old_(t) = Mi e^−k^_v_^t^

Anabolic phase
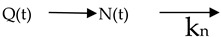

with Q(t) > 0, k_n_ > 0

initial conditions:
N(0) = 0;
final conditions:N(t→∞) = M_f_
the model is described by the following equation:dN(t)/dt = −k_n_ N(t) + Q(t)
how transformed is:s n + k_n_ n = q/s; n = q/s (s + k_n_);
we get A and B from the identity:q/s (s + k_n_) = A/s + B/(s + k_n_); A = q/k_n_; B = −q/k_n_
leading to the following solution:N(t) = (Q/k_n_) (1 − e^−knt^)
that because of
t→∞, N(t→∞) = M_f_,
can be written as:M_new_(t) = M_f_ (1 − e^−k^_n_^t^)


Redistributive phase


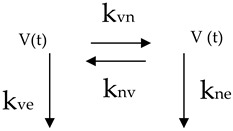


with: k_vn_, k_nv_, k_ve_, k_ve_ > 0with: **k_e_** = k_ve_ = k_ne_; **k_v_** = k_vn_ + k_ve_; **k_n_** = k_nv_ + k_ne_where: k_v_ − k_vn_ = k_n_ − k_nv_ and k_v_ + k_nv_ = k_n_ + k_vn_ = **K**


And since the transition phase is intermediate between the initial phase and the final phase, we must bear in mind that for 0 < t <∞, V (0 < t <∞) = Vx yN (0 < t <∞) = Nx.

The system of differential equations that the model describes is:dV(t)/dt = −k_v_V(t) + k_nv_N(t)
dN(t)/dt = −k_n_N(t) + k_vn_V(t)
with the Laplace-transform:s v + k_v_ v − k_nv_ n = V_x_
s n + k_n_ n − k_vn_ v = N_x_
(s + k_v_) v − k_nv_ n = V_x_
−k_vn_ v + (s + k_n_) n = N_x_

vnKnown(s + k_v_)−k_nv_V_x_−k_vn_(s + k_n_)N_x_D_t_ = (s + k_v_) (s + k_n_) − k_nv_ k_vn_ = (s + α) (s + β); (the roots of the equation in s^2^ are α, β)D_v_ = V_x_ (s + k_n_) + N_x_ k_nv_D_n_ = N_x_ (s + k_v_) + V_x_ k_vn_
where:v = V_x_ (s + k_n_)/(s + α)(s + β) + N_x_ k_nv_/(s + α)(s + β)
n = N_x_ (s + k_v_)/(s + α)(s + β) + V_x_ k_vn_/(s + α)(s + β)

Since the writing of the two previous equations is analogous, it is enough to find the constants A, B, C, F of the first two identities (only of the first equation because those of the second are analogous):V_x_ (s + k_n_)/(s + α)(s + β) = A/(s + α) + B/(s + β)
N_x_ k_nv/_(s + α)(s + β) = C/(s + α) + F/(s + β)
V_x_ (s + k_n_) = A (s + β) + B (s + α) = s(A + B) + A β + B α
where:A + B = V_x_,   and   V_x_ k_n_ = A β + B α
A = V_x_ − B
so what:V_x_ k_n_ = V_x_ β − B β + B α = V_x_ β − B (β − α)
V_x_ (k_n_ − β) = −B (β − α)
thus
B = −V_x_(k_n_ − β)/(β − α)
and, in consequence:A = V_x_ − B = V_x_ + V_x_ (k_n_ − β)/(β − α)
i.e.,:A = V_x_(k_n_ − α)/(β − α)
N_x_ k_nv_ = C (s + β) + F (s + α) = s (C + F) + C β + F α
where:C + F = 0,   e   N_x_ k_nv_ = C β + F α
C = −F
so what:N_x_ k_nv_ = −F β + F α = −F (β − α);
thus:F = −N_x_ k_nv_/(β − α)
and, in consequence:C = N_x_ k_nv_/(β − α)

The following equations:v = V_x_ (s + k_n_)/(s + α)(s + β) + N_x_ k_nv_/(s + α)(s + β)
n = N_x_ (s + k_v_)/(s + α)(s + β) + V_x_ k_vn_/(s + α)(s + β)
can be written:
$$\begin{array}{c} V\left(t\right)=\frac{\mathrm{Vx}\left(\mathrm{kn}-\alpha \right)}{\left(\beta -\alpha \right)\left(s+\alpha \right)}-\mathrm{Vx}\left(\mathrm{kn}-\beta \right)/\left(\beta -\alpha \right)\left(s+\beta \right)+\;\mathrm{Nx}\;\mathrm{knv}/\left(\beta \;-\alpha \right)\left(s+\alpha \right)-\mathrm{Nx}\;\mathrm{knv}/\left(\beta -\alpha \right)\left(s+\beta \right)\\ N\left(t\right)=Nx\left(kv-\alpha \right)/\left(\beta –\alpha \right)\left(s+\alpha \right)–Nx\left(kv-\beta \right)/\left(\beta –\alpha \right)\left(s+\beta \right)+Vx\;\mathrm{kvn}/\left(\beta –\alpha \right)\left(s+\alpha \right)-Vx\;\mathrm{kvn}/\left(\beta –\alpha \right)\left(s+\beta \right)\; \end{array}$$

Of which, we obtain:**V(t) =** V_x_(k_n_ − α)e^−^^αt^/(β − α) − V_x_(k_n_ − β)e^−^^βt^/(β − α) + N_x_ k_nv_ e^−^^αt^/(β − α) − N_x_ k_nv_ e^−^^βt^/(β − α)
**N(t) =** N_x_(k_v_ − α)e^−^^αt^/(β − α) − N_x_(k_v_ − β)e^−^^βt^/(β − α) + V_x_ k_vn_ e^−^^αt^/(β − α) − V_x_ k_vn_ e^−^^βt^/(β − α)

Given that
M(t) = V(t) + N(t)
then:M(t) = V_x_(k_n_ − α + k_vn_)e^−^^αt^/(β − α) + N_x_(k_v_ − α + k_nv_)e^−^^αt^/(β − α)+
        −V_x_(k_n_ − β + k_vn_)e^−^^βt^/(β − α) − N_x_(k_v_ − β + k_nv_)e^−^^βt^/(β − α)
M(t) = [V_x_(k_n_ − α + k_vn_) + N_x_(k_v_ − α + k_nv_)] e^−^^αt^/(β − α)+
     −[V_x_(k_n_ − β + k_vn_) + N_x_(k_v_ − β + k_nv_) ] e^−^^βt^/(β − α)
and remembering that
 k_e_ = k_v_ − k_vn_ = k_n_ − k_vn_; k_v_ + k_vn_ = k_n_ + k_vn_ = K
then:M(t) = [V_x_(K − α) + N_x_(K − α)] e^−^^αt^/(β − α) − [V_x_(K − β) + N_x_(K − β)] e^−^^βt^/(β − α)
M_redistributive_ (t) = (V_x_ + N_x_)(K − α) e^−^^αt^/(β − α) − (V_x_ + N_x_)(K − β) e^−^^βt^/(β − α)

Finally how (K − α) ~ (K − β) ~ (β − α), can be approximated:M_redistributive_(t) = (M_i_ + M_f_) (e^−αt^ − e^−βt^)
that in the initial and final conditions:for t = 0 M_redistributive_ (0) = 0
and for t→∞ M_redistributive_ (t→∞) = 0

Finally, the solution of the model is given by the sum of the three solving equations of the catabolic, anabolic, and distributive processes:M_c_(t) = M_i_ e^−kv^^t^ + M_f_ (1 − e^−kn^^t^) + (M_i_ + M_f_)(e^−^^α^^t^ − e^−^^β^^t^)

Since the metabolic rate constant of the “old” mass is substantially the initial metabolic constant and that of the “new” mass is defined as the final or final, the above equation becomes:M_c_(t) = M_i_ e^−^^kmi^^t^ + M_f_ (1 − e^−^^kmf^^t^) + (M_i_ + M_f_)(e^−^^α^^t^ − e^−^^β^^t^**)**
where M_c_ is the body mass at time t, M_i_ the body mass at the beginning of the observations, M_f_ the body mass at the end, respectively, and α and β are the constants controlling the distributive process. To facilitate the calculations necessary to obtain the numerical solution of the model on the experimental data set, specially compiled software (Dies4) was used. Body mass values were measured monthly by means of an electronic balance (sensitivity of 0.1 kg). The metabolic rate constants were calculated with the following formula:k_m_ [h^−1^] = −(1/24) ln [(M_c_ − M_a_)/M_c_]
where M_c_ (kg) is the body mass and M_a_ (kg) the daily food mass.

If this equation is applied to all the pairs of values for M_C_ and M_a_, the metabolic constants of the body mass can be calculated at any time. A balanced diet was assigned for each of the patients with the aim of making the patients tend toward normal with respect to their BMI interval. Body mass was measured initially and then subsequently at each control. The initial food mass was reported by the patients through the food diary and subsequently replaced by the dietary mass of the assigned diet. Patients were required to follow the diet scrupulously and to maintain the same lifestyle at least as long as necessary to carry out the checks. For each of the patients, the metabolic constants were calculated during the control visits using the same formula with which the initial values were calculated. Therefore, the constants varied because the body mass, not the food mass, varied over time. The metabolic constants are stable when body mass stabilizes. Considering body mass as a result of the three catabolic, anabolic, and redistributive phases for each patient, a temporal trend was constructed by inserting the initial, intermediate, and final constants into the formal solution of the model. The expected body masses were calculated, whereas the observed values were measured experimentally during the control visits. At every time on the curve, each patient was characterized by a couple of expected–observed values. A database was constructed by randomly inserting the following parameters for each patient: sex, age, initial body mass, body mass index (BMI), initial k_m_, final body mass, final k_m_, number of control visits, deviance, and variance. Figure 1 shows an example of fitting a patient’s body mass.

Deviance is the sum of the squares of the differences between the calculated (expected) data and the experimentally measured (observed) data: Dev = $\sum_{1}^{n}\;$= (Y_observed_ − Y_expected_)^2^. The variance is Var = $\frac{\mathrm{Dev}}{\mathrm{DF}}$, where Dev = deviance and DF = degrees of freedom. Appendix A shows all values obtained for the studied subjects and Appendix A reflects all values with statistical analysis.

## 4. Discussion

The logical sequence of the phases that lead to the realization of a biomathematic model such as the one proposed in this research are: (i) draw one or more block diagrams that have biological sense and, above all, indicate the connections and direction of matter transfers, (ii) describe the scheme with a system of differential equations (Formal model), (iii) solve the system of differential equations to obtain the integral equation (Formal solution of the model), (iv) calculate, from the formal solution, the values that the dependent variable (in our case the Body Mass) takes in correspondence with the independent variable (time), (v) analyze all pairs of values calculated and measured experimentally to understand if there are significant differences (validation process) and (vi) the model is valid to interpret the experimental data only if the variance, calculated on the total set of value pairs and/or on the subgroups tends to zero (at 240 degrees of freedom it is sufficient that it is less than 1).

All mean variances of the pathophysiological groups are less than 1 and are no different from the total variance. The model is validated. However, if we compare the mean variances of the individual pathological groups with the group of healthy people, it is observed that the model applied to the groups of anorexia and tumors (much more evident in anorexia) proves to be valid but at the limit of significance. This is due to the insufficient number of samples. A more in-depth investigation, which could remove any doubt about the validity of the model also in these two groups, is desirable and to be carried out in future research. In fact, the variance analysis carried out on the results showed that no significant differences existed between the expected and the experimental data; therefore, we concluded that the model was suitable for interpreting trends in body mass over time. In our viewpoint, the limits of this study are determined by the purpose of the research itself and therefore by the consequent experimental plan. No other result, obtained outside the experimental plan, can be considered valid. If by limits we mean the critical elements then it can be said that the random choice of the sample guarantees the statistical representativeness of the population but cannot guarantee in the same way the comparison between those groups, internal to the sample, which have lower degrees of freedom. For example, in this study, the comparison between anorexic and oncological groups with healthy people could be questioned for the low number of degrees of freedom at stake (37 and 43 respectively). On the other hand, it should be considered that the analysis of variance on partial sets and on the whole set cannot be misunderstood. We are not demonstrating the equality between pathophysiological situations, which obviously does not exist and cannot be there. Instead, we are evaluating whether, in each pathophysiological situation, the model can interpret the experimental data.

In the future, the research could be extended to each single metabolic compartment (protein, aqueous, glucose, and lipid). Metabolic constants are specific to the subject and are the result of the functional state of the subject, and do not depend on whether they belong to a particular pathophysiological class. Our results confirm that body mass and its variation do not depend on energy but on the metabolic speed constants [19] where the body mass of an individual is determined by mass balance, regulated by corresponding metabolic rate, calculated by the BFMNU method, thanks to which the macronutrients in the diet are absorbed, redistributed and eliminated. In fact, this study reflected that a significant correlation, although not straight, is demonstrated between ∆% of food energy, supplied after processing through the dietary BFMNU method, and the ∆% of body mass, obtained following the dietetic path. Furthermore, this tool was applied in university students [20]. Currently, the clinical application of predictive equations [2,3,4,5,6,7,8,9,10] is useful for calculating the resting metabolic rate. However, the best fit allows us to predict body mass instantaneously as well as throughout life, providing the significant advantage of control over the whole process.

Currently, the diet methodology is premised on energy and tends to understand body mass as a result of dietary energy contained in the diet. It is said that body mass is related to introduced food energy, but the correlation between the two is so low as to be considered casual and not significant. Within the same patient, starting from a steady state, the only method to determine the end point is to evaluate the percentage variation of the energy contained in the assigned diet with respect to the previous diet. This is correlated, but not in a straight line, to the percentage change in the body mass [5]. The overall process that leads to the formation of body mass is the sum of partial dynamic processes: catabolic, anabolic, and distributive. These processes are contemporary and occur throughout life, even if at different rates depending on age and pathophysiological conditions. The modelling approach allows us to determine the speed constants that regulate these three processes. Body mass depends on the diet mass and mainly on the speed constants (k) with which the mass is transformed, which depends on metabolism [7]. The calculation of speed constants indicates the space–time context of the biological process and predicts its progress and possible outcomes. Therefore, if nutritional needs can be determined on the basis of metabolic constants, then optimal and specific diets can be formulated. The main purpose of the method is to optimize health with proper nutrition, which means finding a diet that is able to produce a combined change in body mass and composition and provide useful energy to the body to improve its performance [8]. Several authors have examined the importance of other factors on resting metabolic rate and energy expenditure, including specific behaviors [21] and environmental factors [22,23]. The literature indicates that measured RMR (mRMR) is positively correlated with BMI, total fat mass, and fat-free mass [24,25,26]. In longitudinal studies, animal models were used to understand and examine individual variation in resting metabolic rate. Krom et al. [27] used model components from obesogenic environments when using animal models to elucidate aspects of human obesity. The advantage of animal models is that rats can be selected, placed in a well-controlled environment, and followed throughout the development, treatment, and relapse of nutritional diseases [28,29,30,31]. Unlike humans, the behavior of rats is not influenced by peer pressure, the concept of an ideal physique, or other societal factors that provide the motivation to alter behavior [28]. Recently, the applicability of BFMNU with the BMI has been used to define a new formula called BMI-BFMNU where it is possible to obtain an indication of the body structure related to the amount of fat [32].

## 5. Conclusions

In summary, the use of BFMNU is useful to systemically determine the metabolic processes of mass and energy and to overcome the many contradictions of the methods already in use. It is a phenomenological and mathematical method applicable to any individual in any pathophysiological condition, and its primary objective is to determine the optimal diet for producing a synergistic change in body mass, body composition, and energy to improve the functionality of the body with the smallest possible variation in comparison to previous eating habits. The promotion of this study reflected that the BFMNU method is a model that can help describe trends in body mass and its variations over time using a continuous function that is useful in daily clinical work. Together with the advantages, however, it is necessary to highlight some limitations that derive precisely from the very nature of the method. The collection of data, information, direct measurements, processing and calculations takes time on the part of specialists.

## Figures and Tables

**Figure 1 nutrients-14-03575-f001:**
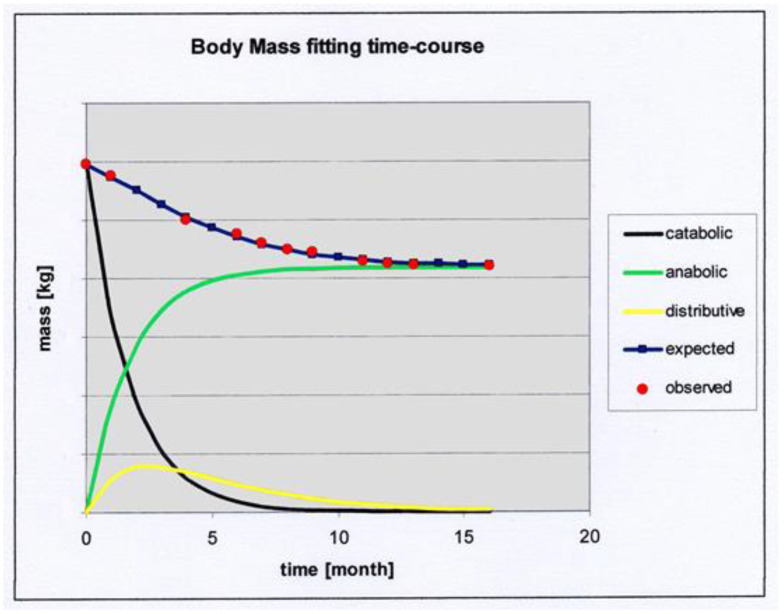
An example of fitting a patient’s body mass. The expected body mass (blue line and points) is the sum of three contributions from the catabolic (dark line), anabolic (green line), and redistributive phases (yellow line).

## Data Availability

Not applicable.

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
