# Peer review of "Proposal of a Mathematical Model to Monitor Body Mass over Time in Subjects on a Diet"

_nutrients, 2022, doi:10.3390/nu14173575_

Round 1

Reviewer 1 Report

In “Proposal of a mathematical model to monitor body mass over time in subjects on a diet”, the authors propose a mathematical model that explains trends in body mass and its variations using a continuous function, valid in any physiological and pathological condition. For this manuscript, there are some recommendations to consider as detailed below.

1. The author proposes a good mathematical model, but the formula in the results section is a little messy, making it difficult for readers to understand.

2. The picture in Figure 1 is not clear enough and lacks the value of the ordinate. It is recommended to modify Figure 1.

3. The author mentions that the mathematical model is valid in many physiological and pathological condition. It is recommended to add a verification part, which can be displayed in the form of figures or tables.

4. The authors used analysis of variance to demonstrate that there was no significant difference between the expected and experimental data and did not present the results of the analysis of variance in the text,which can be displayed in the form of figures or tables.

Author Response

Reviewer’s comment: In “Proposal of a mathematical model to monitor body mass over time in subjects on a diet”, the authors propose a mathematical model that explains trends in body mass and its variations using a continuous function, valid in any physiological and pathological condition. For this manuscript, there are some recommendations to consider as detailed below.

The author proposes a good mathematical model, but the formula in the results section is a little messy, making it difficult for readers to understand.

Author’s comment: According to reviewer’s comment, authors have been modified formula to clarify the result section.

Reviewer’s comment: The picture in Figure 1 is not clear enough and lacks the value of the ordinate. It is recommended to modify Figure 1.

Author’s comment: According to reviewer’s comment, authors have added explanation in the figure caption to clarify the sense of the Figure.

Reviewer’s comment: The author mentions that the mathematical model is valid in many physiological and pathological condition. It is recommended to add a verification part, which can be displayed in the form of figures or tables.

Author’s comment: Figure 1 demonstrated an example of one subject of this study and figure caption has been added to clarify it.

Reviewer’s comment: The authors used analysis of variance to demonstrate that there was no significant difference between the expected and experimental data and did not present the results of the analysis of variance in the text,which can be displayed in the form of figures or tables.

 Author’s comment: The analysis of variance allows us to understand if there are significant differences between the Mass values calculated with the formal solution of the model (equation on line 182 of the text) and those measured directly on patients. This analysis can be done both on the overall set of data, and on the partial sets of individual patients and groups (healthy, pathological, etc.). Regardless of the fact that the pathological groups formed at random (they were not chosen on purpose because they did not fit into the experimental plan), the analysis of variance showed that there is no significant difference between the calculated mass values and those measured experimentally.

Acccording to this comment, authors have been added in supplementary material other table in Excel with all information to be verified by this reviewer or any reader.

Reviewer 2 Report

In this study, the authors raised a mathematical model to monitor body mass over time in subjects on a diet. I have several concerns about this study.

1. In the abstract, the authors need to give a background for why you performed this research in addition to the limitations of recent studies.

2. The significance of this study should be mentioned in the abstract. 

3. More information is needed in the Introduction, such as whether similar research was performed in the past few years, what the benefits of this model is for the development of public health. 

4. How did you select the participants in the study?

5. In the section of "Meterials and Methods", why did you think "The number of experimental samples was adequate"? How did you calculate the number of samples? 

6. What were the limitations of this study?

7. How did the model in this study promote public health?

8. The discussion is too simple. More information related to this model should be added to the discussion. 

9. Moderate English editing are required. 

Author Response

Reviewer’s comment: In this study, the authors raised a mathematical model to monitor body mass over time in subjects on a diet. I have several concerns about this study.

In the abstract, the authors need to give a background for why you performed this research in addition to the limitations of recent studies.

Author’s comment: The reason for the research is to explain the temporal trend of body mass and its   variations of a large and heterogeneous sample of volunteers who have undergone a change in their eating habits (new diet). According to reviewer’s comment, authors have included this part in abstract section.

Reviewer’s comment:  The significance of this study should be mentioned in the abstract.

Author’s comment: According to reviewer’s comment, authors have included this part in abstract section.

Reviewer’s comment: More information is needed in the Introduction, such as whether similar research was performed in the past few years, what the benefits of this model is for the development of public health.

Author’s comment: According to this comment, more information have been added in the introduction section.

Reviewer’s comment: How did you select the participants in the study?

Author’s comment: Participants in the study were selected in the University laboratory being the selection the guarantee to finish the study.

Reviewer’s comment: In the section of "Meterials and Methods", why did you think "The number of experimental samples was adequate"? How did you calculate the number of samples?

Author’s comment: In order to obtain a dataset suitable for the analysis of variance, only subjects who had performed no less than four direct measurements of body mass over the entire transition period of initial body mass to the final one were admitted to the study. The size of the sample thus selected largely guaranteed the applicability of the statistical analysis.

Reviewer’s comment: What were the limitations of this study?

Author’s comment: The limits of the study are determined by the purpose of the research itself and therefore by the consequent experimental plan. No other result, obtained outside the experimental plan, can be considered valid.  It has been added in the manuscript.

Reviewer’s comment: How did the model in this study promote public health?

Author’s comment: If you can find a valid model to interpret the trend of mass over time, you can also think that the same model can be used to predict the mass value achievable through a new diet. And if this is possible it is also easy to imagine the relapse of the positive effects on public health.

Reviewer’s comment: The discussion is too simple. More information related to this model should be added to the discussion.

Author’s comment: According to reviewer’s comment, it has been added in the discussion section.

Reviewer’s comment: Moderate English editing are required.

Author’s comment: According to reviewers’ comments and re-writen, we think that the manuscript is adequate.

Reviewer 3 Report

The authors conducted a study proposal of a mathematical model to monitor body mass over time in subjects on a diet. There are insufficient experimental procedures and data using a function of time for absorption, metabolism, and excretion, which are metabolic processes for the diet.

Author Response

Reviewer’s comment: The authors conducted a study proposal of a mathematical model to monitor body mass over time in subjects on a diet. There are insufficient experimental procedures and data using a function of time for absorption, metabolism, and excretion, which are metabolic processes for the diet.

Author’s comment: According to this comment and other reviewer’s comments, authors have been re-written to clarify the manuscript.

Round 2

Reviewer 1 Report

All the comments have been revised, I have no new suggestion.

Author Response

Reviewer’s comment: All the comments have been revised. I have no new suggestion.

Author’s comment: Thank you for your comment.

Reviewer 2 Report

1. Background is not aim. The authors have added the aim of this study to the abstract. However, the background of this research is limited. 

2. Is there any similar research performed in the past few years? This information is important and should be added in the Introduction.

3. What was the criteria of the participants involved in the research? Specific methods are needed. 

4. Why did you think "The number of experimental samples was adequate"? How did you calculate the number of samples? 

5. What were the limitations of this study? It should be added in the discussion.

6. The promotion of this study for this study should be added in the conclusion.

7. The discussion is too simple. Recent results in this study should not need to be repeated in the discussion. More information related to this model should be added.

8. Moderate English editing are required.

Author Response

Reviewer’s comment: Background is not aim. The authors have added the aim of this study to the abstract. However, the background of this research is limited.

Author’s comment: According to reviewer’s comment, we have re-written abstract section.

Reviewer’s comment: Is there any similar research performed in the past few years? This information is important and should be added in the Introduction.

Author’s comment: The authors do not have similar research done on the basis of a biomathematic model such as that described in this article. This is the first article published with this model in the world literaure.

Reviewer’s comment: What was the criterio of the participants involved in the research? Specific methods are needed.

Author’s comment: The criterion was to motivate them to voluntarily join the research first of all to take advantage of the benefits of the new diet protocol. And to collaborate to improve the health service of prevention and treatment through nutrition. It has been added in the material and methods section.

Reviewer’s comment: Why did you think “The number of experimental simples was adequate”? How did you claculate the number of samples?

Author’s comment: The distribution of variance indicates that a set of data pairs that has at least 240 degrees of freedom is congruous. Since only volunteers who had carried out at least 4 experimental checks would have been admitted to the study, 60 patients would have been enough. It has been added in the materials and method section from lines 73 to 80.

Reviewer’s comment: What were the limitations of this study?It should be added in the discussion.

Author’s comment: The limits of the study are determined by the purpose of the research itself and therefore by the consequent experimental plan. No other result, obtained outside the experimental plan, can be considered valid.

If by limits we mean the critical elements then it can be said that the random choice of the sample guarantees the statistical representativeness of the population but cannot guarantee in the same way the comparison between those groups, internal to the sample, which have lower degrees of freedom. For example, in this study, the comparison between anorexic and oncological groups with healthy people could be questioned for the low number of degrees of freedom at stake (37 and 43 respectively). On the other hand, it should be considered that the analysis of variance on partial sets and on the whole set cannot be misunderstood. We are not demonstrating the equality between pathophysiological situations, which obviously does not exist and cannot be there.  Instead, we are evaluating whether, in each pathophysiological situation, the model can interpret the experimental data.

Reviewer’s comment: The promotion of this study for this study should be in the conclussion.   

Author’s comment: According to reviewer’s comment, authors have been added in the conclussion section.

Reviewer’s comment: The discussion is too simple. Recent results in this study should not need to be repeated in the discussion. More information related to this model should be added.

Author’s comment: According to reviewer’s comment, authors have expanded the discussion section. In our viewpoint, we think that it help to clarify the manuscript. Furthermore, we have included an explanation about this biomathematic model.

The logical sequence of the phases that lead to the realization of a biomathematic model such as the one proposed in this research are:

a- Draw one or more block diagrams that have biological sense and, above all, indicate the connections and direction of matter transfers

b- Describe the scheme with a system of differential equations (Formal model)

c- Solve the system of differential equations to obtain the integral equation (Formal solution of the model).

d- Calculate, from the formal solution, the values that the dependent variable (in our case the Body Mass) takes in correspondence with the independent variable (time).

     e-  Analyze all pairs of values calculated and measured experimentally to understand if there are significant differences (validation process).

f-            The model is valid to interpret the experimental data only if the variance, calculated on the total set of value pairs and / or on the subgroups tends to zero. (at 240 degrees of freedom it is sufficient that it is less than 1).

All mean variances of the pathophysiological groups are less than 1 and are no different from the total variance. The model is validated. However, if we compare the mean variances of the individual pathological groups with the group of healthy people, it is observed that the model applied to the groups of anorexia and tumors (much more evident in anorexia) proves to be valid but at the limit of significance. This is due to the insufficient number of the sample. A more in-depth investigation, which could remove any doubt about the validity of the model also in these two groups, is desirable and to be carried out in a future research.

Reviewer’s comment: Moderate English editing are required.

Author’s comment: According to reviewers’ comments and re-writen, we think that the manuscript is adequate.

Reviewer 3 Report

The reviewers' requests were fulfilled, and the data corrections were successful.